# MATEY: MULTISCALE ADAPTIVE FOUNDATION MODELS FOR SPATIOTEMPORAL PHYSICAL SYSTEMS

## ABSTRACT

Accurate representation of the multiscale features in spatiotemporal physical systems using vision transformer (ViT) architectures requires extremely long, computationally prohibitive token sequences. To address this issue, we propose an adaptive tokenization scheme which dynamically adjusts the token sizes based on local features. Moreover, we present a set of spatiotemporal attention schemes, where the temporal or axial spatial dimensions are decoupled, and evaluate their computational and data efficiencies. We assess the performance of the proposed multiscale adaptive model, MATEY, in a sequence of experiments. The results show that adaptive tokenization achieves improved accuracy without significantly increasing token sequence length, but the improvement deteriorates in more complex data configurations. Compared to a full spatiotemporal attention scheme or a scheme that decouples only the temporal dimension, we find that fully decoupled axial attention is less efficient and expressive, requiring more training time and model weights to achieve the same accuracy. Finally, we demonstrate in two fine-tuning tasks featuring different physics that models pretrained on PDEBench data outperform the ones trained from scratch, especially in the low data regime with frozen attention.

## 1 INTRODUCTION

Developing foundation models for physical systems is vital for energy generation, earth sciences, and power and propulsion systems. These models offer faster solutions than physics-based simulations and can generalize better across multiple systems than single-purpose AI approaches. However, their application to physical systems, often characterized by multiple sub-processes at different scales, is still in the early stages. For instance, fluid flowing around a cylinder creates a von Kármán vortex street, a highly dynamic flow with rapidly evolving vortices. Accurate solutions of such multiscale systems requires a very high resolution representation to capture the most complex features across space and time. However, for scientific machine learning as for modeling and simulation, using very high resolutions to achieve accurate solutions incurs significant computational cost. This is particularly true for developing foundation models using vision transformer (ViT)-based architectures, as using the standard self-attention mechanism for extremely long spatiotemporal sequences can become prohibitively computationally expensive.

Efficient representation of multiscale features in high-resolution inputs has been an active research topic in computer vision. Three broad approaches can be characterized. First, multiscale models like Swin Transformer (Liu et al., 2021) and MViTv2 (Li et al., 2022) introduce multiple stages with decreasing resolution and increasing feature dimension for efficient hierarchical representations. Second, computational techniques have been developed which facilitate training on long sequences (e.g., sequence parallelism across GPUs (Jacobs et al., 2023)) or reduce the effective sequence length in the attention kernel (e.g., decomposing attention along axial directions (Ho et al., 2019)). Third, the actual sequence length can be directly shortened by pruning and merging tokens ((Haurum et al., 2023; Meng et al., 2022; Yin et al., 2022; Bolya & Hoffman, 2023)), though this strategy may lead to critical information loss (Liu et al., 2024).

These techniques have recently been adopted in scientific machine learning (sciML) for physical systems. For example, the atmosphere foundation model Aurora (Bodnar et al., 2024) uses Swin Transformer, while axial attention is applied by MPP (McCabe et al., 2023). Despite the progress,

computational constraints remain a bottleneck, as existing approaches do not yet handle high-fidelity solutions of applications such as computational fluid dynamics, in which input sequences can easily exceed billions of tokens. More efficient algorithms are needed to enable the development of foundation models for multiscale multiphysics systems.

In this work, we develop a multiscale adaptive foundation model, MATEY (see Figure 1), that provides two key algorithmic contributions to address the challenges posed by spatiotemporal physical systems. First, inspired by the adaptive mesh refinement (AMR) technique, we introduce an adaptive tokenization method that dynamically adjusts patch sizes across the system based on local features, which provides as much as a $2\times$ reduction in compute for similar or higher accuracy. Second, we present a set of spatiotemporal attention schemes based on the axial attention (Ho et al., 2019) that differ in their decomposition of long spatiotemporal sequences and identify the cost in time-to-accuracy for axial attention. Finally, we assess the fine-tuning performance of models pretrained on PDEBench (Takamoto et al., 2022) in two highly out-of-distribution settings, colliding thermals and magnetohydrodynamics (MHD), that include additional physical variables not included in pretraining and observe the pretrained models outperforming random initialized models.

## 2  RELATED WORK

**Scientific foundation models.**  Several research directions have been explored for building foundation models for physical systems, including multiple physics pretraining (McCabe et al., 2023) with PDEBench data, input augmentation with PDE system configurations (Hang et al., 2024), robust pretraining schemes (Hao et al., 2024), fine-tuning effectiveness investigations (Subramanian et al., 2024), and data-efficient multiscale ViT architectures (Herde et al., 2024). While these work made remarkable progress, they do not directly address the issue of token sequence length, which becomes a computation bottleneck when applying ViTs to high dimension or high resolution data.

**Multiscale ViTs.**  While most multiscale ViTs achieve hierarchical representations via multi-stage attention blocks at different resolutions, e.g., MViTv2 (Li et al., 2022) and Swin Transformer (Liu et al., 2021), there are a few focusing on tokenization schemes, e.g., (Yin et al., 2022; Fan et al., 2024; Zhang et al., 2024; Havtorn et al., 2023). Among these, the single-stage MSViT with dynamic mixed-scale tokenization (Havtorn et al., 2023), which leverages a learnable gating neural network for token refining, is most related to our work. This approach requires a tailored gate loss function and an adaptive trimming scheme to handle the high overhead cost, which in return hurts gate training accuracy. In contrast, the tokenization scheme in MATEY adaptively adjusts the patch sizes directly based on local feature scales, which is simpler and more direct.

**Axial attentions.**  The quadratic scaling nature of attention makes it computationally prohibitive for extremely long token sequences from multidimensional systems. To address this challenge, (Ho et al., 2019) proposed the axial attention, which decomposes the full attention into a sequence of attention operations along each axis. It reduces the attention cost from $\mathcal{O}(N^{2d})$ to $\mathcal{O}(N^{d+1})$, for a given $d$-dimensional system with $N^d$ tokens. ViViT (Arnab et al., 2021) factorized the spatiotemporal attention into spatial- and temporal-dimensions for video classification. (McCabe et al., 2023) applied the axial attention in the Axial ViT (AViT) for spatiotemporal solutions of physical systems. While these spatiotemporal attention schemes can reduce the sequence length and hence the attention cost, their impact on accuracy in physical systems is unclear.

## 3  MATEY, EXPLAINED

We propose multiscale adaptive foundation models, MATEY, to predict two-dimensional spatiotemporal solutions of multiple physical systems. The architecture of MATEY is illustrated in Figure 1. Given a sequence of $T$ past solutions of some physical system at time $t$, MATEY predicts the solution at a future time $t + t_{\text{lead}}$ by learning from sequences of solutions for multiple physical systems. Specifically, MATEY learns a model $\mathbf{f_w}$ such that $\mathbf{u}_{t+t_{\text{lead}}} \approx \mathbf{f_w}(\mathbf{u}_{t-T+1}, \ldots, \mathbf{u}_t; t_{\text{lead}})$ by training parameters $\mathbf{w}$ to minimize the loss of the prediction from the solution sequence $\mathbf{U} = [\mathbf{u}_{t-T+1}, \ldots, \mathbf{u}_t]$ against the future solution with a lead time $\mathbf{u}_{t+t_{\text{lead}}}$. In the following paragraphs, we give detailed descriptions for each component in MATEY.

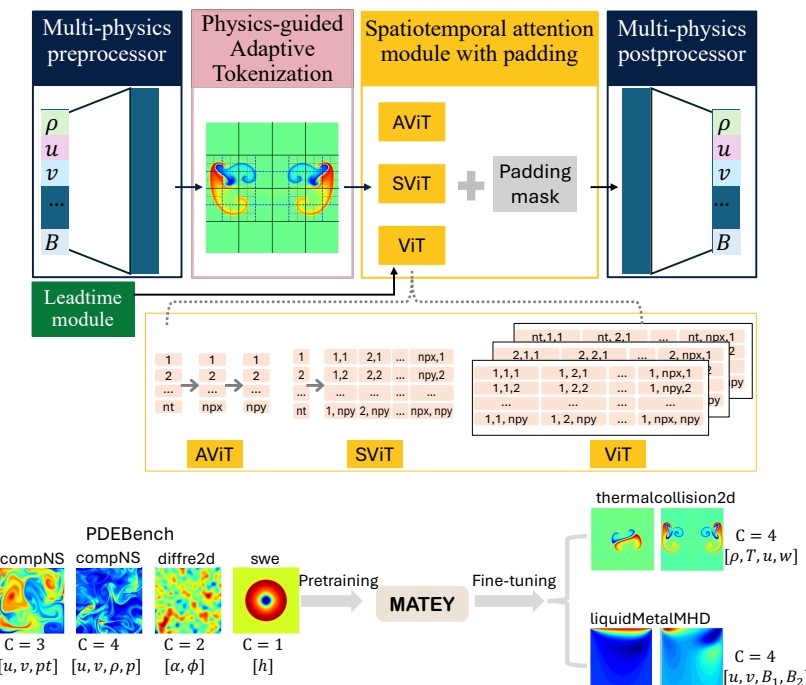

Figure 1: MATEY: multiscale adaptive foundation models for spatiotemporal physical systems.

**Multi-physics preprocessor, postprocessor, and training.** To accommodate multiple physical systems with different sets of variables at different spatial resolutions, we adopt the multi-physics preprocessor and postprocessor used in the MPP work (McCabe et al., 2023). For system $k$ with $C_k$ variables, the preprocessor first embeds solutions $\mathbf{u}_t(x,y) \in \mathbb{R}^{C_k}$ to a latent space $\mathbb{R}^{C_{\text{uni}}}$, where $C_{\text{uni}} \gg C_k$ is shared among all systems. The resulting embedded solution passes through a convolutional block in the tokenization module and is converted into patch sequences in $\mathbb{R}^{C_{\text{emb}}}$, which are further passed to the attention block and then mapped back to $\mathbb{R}^{C_k}$ by the postprocessor, to predict the solution fields of the $C_k$ variables. To handle solutions with different resolutions, we follow the approach in MPP by performing system-based sampling in the training process and fusing information from samples from different systems via gradient accumulation. We employ a convolutional neural network (CNN) in the tokenization module and 2D transposed convolutional blocks in the multi-physics postprocessor to convert between patch sequences and spatiotemporal solution fields. The CNN block performs the conversion of the embedded solutions in the latent space $\mathbb{R}^{C_{\text{uni}}}$ to the patch sequences in $\mathbb{R}^{C_{\text{emb}}}$, while the transposed convolutional module converts the sequences back to the solution fields. Specifically, the preprocessor embeds the solution of system $k$, $\mathbf{U}_k \in \mathbb{R}^{T \times H \times W \times C_k}$ into the unified latent representation $\mathbf{U} \in \mathbb{R}^{T \times H \times W \times C_{\text{uni}}}$, which is tokenized into sequences $\mathbf{Z}_p^0 \in \mathbb{R}^{nt \times npx \times npy \times C_{\text{emb}}}$, where $nt = T/p_t$, $npx = H/p_x$, and $npy = W/p_y$ with prescribed patch size $[p_t, p_x, p_y]$ in the temporal and spatial dimensions. On the other hand, the postprocessor decodes the attention output $\mathbf{Z}_p^L \in \mathbb{R}^{npx \times npy \times C_{\text{emb}}}$ to the prediction $\mathbf{u}_{\text{pred}} \in \mathbb{R}^{H \times W \times C_k}$ for system $k$. In our work, we keep $p_t = 1$ and $C_{\text{uni}} = C_{\text{emb}}/4$.

**Attention mechanisms — AViT, SViT, and ViT.** The standard ViT attention mechanism takes into account the attention across the entire set of spatiotemporal dimensions, which results in a high attention cost when extremely long spatiotemporal token sequences (e.g., from high-resolution spatiotemporal data) are considered. To address this issue, various factorized attention mechanisms have been proposed, such as AViT (Ho et al., 2019; McCabe et al., 2023) and a spatio-temporal decoupled attention (Arnab et al., 2021), referred to as SViT here. These attention mechanisms mainly consist of the same multihead self attention (MHSA) and feed forward multi-layer perceptron (MLP) but differ in their attention block architecture. When $L$ attention blocks are cascaded, the

standard attention block in ViT is given as

$$
\begin{aligned}
\widehat{\boldsymbol{Z}}_p^0 &= [\boldsymbol{z}_1^0, \boldsymbol{z}_2^0, \dots, \boldsymbol{z}_N^0] + \boldsymbol{E}_{\text{pos}}, \\
\boldsymbol{Z}_p^1 &= \text{MLP}(\widetilde{\boldsymbol{Z}}_p^1) + \widetilde{\boldsymbol{Z}}_p^1, \qquad \widetilde{\boldsymbol{Z}}_p^1 = \text{MHSA}(\widehat{\boldsymbol{Z}}_p^0) + \widehat{\boldsymbol{Z}}_p^0 + \text{MLP}(t_{\text{lead}}), \\
\boldsymbol{Z}_p^\ell &= \text{MLP}(\widetilde{\boldsymbol{Z}}_p^\ell) + \widetilde{\boldsymbol{Z}}_p^\ell, \qquad \widetilde{\boldsymbol{Z}}_p^\ell = \text{MHSA}(\boldsymbol{Z}_p^{\ell-1}) + \boldsymbol{Z}_p^{\ell-1}, \qquad \ell = 2, \dots, L
\end{aligned}
\tag{1}
$$

where $[\boldsymbol{z}_1^0, \dots, \boldsymbol{z}_N^0]$ denotes the full spatiotemporal token sequence of length $N$ with each token $\boldsymbol{z}_i^0 \in \mathbb{R}^{C_{\text{emb}}}$, $\boldsymbol{E}_{\text{pos}}$ is a positional embedding term, and each MHSA and MLP is followed by an `InstanceNorm1d` module. In ViT, the token sequence is composed of full spatiotemporal patches, meaning $N = nt \cdot npx \cdot npy$, resulting in an overwhelming costs of $\mathcal{O}((nt \cdot npx \cdot npy)^2)$ operations for attention. In contrast, SViT decouples the attention into $npx \cdot npy$ time-attention blocks and $nt$ space-attention blocks cascaded sequentially, as in "MHSA$_{\text{time}}$ $\rightarrow$ MHSA$_{\text{space}}$ $\rightarrow$ MLP", which reduces the MHSA cost to $npx \cdot npy \cdot \mathcal{O}(nt^2) + nt \cdot \mathcal{O}((npx \cdot npy)^2)$. AViT further decomposes the space-attention in SViT into two axial directions following the same approach, which leads to a cost of $npx \cdot npy \cdot \mathcal{O}(nt^2) + nt \cdot npy \cdot \mathcal{O}(npx^2) + nt \cdot npx \cdot \mathcal{O}(npy^2)$. The decomposition approach taken in both AViT and SViT neglects some spatiotemporal correlations and thus gives shorter token sequence length for each attention blocks, at the cost of introducing additional attention blocks. These extra attention blocks moderately increase the model size, as shown in Table 1. Note that within the same size category, AViT and ViT are larger than ViT due to the additonal MHSA, while AViT and ViT have similar sizes because AViT reuses the same attention for different spatial directions. In MATEY, we implement the three attention mechanisms, AViT, SViT, and ViT, and evaluate their performance on test problems to study how the lost spatiotemporal correlations affect the quality of the solution and to assess the impact of decoupled attentions with additional attention blocks on the learning efficiency for multi-physics foundation models.

**Adaptive tokenization.** Smaller patch sizes are preferred for better representation accuracy, as ViTs can capture long-range correlations between patches well but lack inductive biases within patches. However, features in physical systems often cross multiple length scales and exhibit strong spatiotemporal inhomogeneities. Consequently, constant patch sizes that are small enough to provide good accuracy in the necessary regions of such systems result in impractically long token sequence lengths over the entire domain. To address this issue, we propose an adaptive ViT that dynamically adjusts the tokenization patch sizes according to local physical features. To maximize expressiveness, we start with coarse patching and identify the most complex patches in each sample based on a simple metric, such as the variance of local features. The identified patches are further refined to the sub-token-scale (STS) to improve representation accuracy in these regions. Adaptive patch size leads to patches at varying length across samples, which are handled with padding mask. Patch position and patch area bias are represented following the embedding method in (Bodnar et al., 2024).

For a given solution $\boldsymbol{u}_i \in \mathbb{R}^{H \times W \times C}$ and an initial coarse patch size $[p_{x_1}, p_{y_1}]$, the patch sequence is refined adaptively based on local patch variance with two parameters, $[p_{x_{\text{ref}}}, p_{y_{\text{ref}}}]$ and $\gamma_{\text{ref}}$, as shown in Figure 2. The resulting STS tokens can be incorporated in two ways. In the first approach, referred to as "Adap_Mul" (for adaptive multi-resolution tokenization), we consider the coarse and STS tokens as separate sequences, passing through the attention blocks serially. In the second approach, referred to as "Adap_Mix" (for adaptive mixed-resolution tokenization), we append the sequence of STS tokens directly to the end of the sequence of coarse tokens. While the second approach leads to relatively longer token sequences, it has the potential benefit of better capturing cross-scale correlations than the decoupled first approach.

**Pretraining and fine-tuning.** We pretrain the models on PDEBench data, which include five basic 2D systems: incompressible flows, compressible flows, turbulent flows, reaction-diffusion systems, and shallow water equations. We consider two fine-tuning cases: 1) colliding thermals between a cold and a warm bubbles from MiniWeather simulations (Norman, 2020) and 2) lid-driven cavity MHD flows (Fambri et al., 2023). As discussed in detail in Appendix A.1, these fine-tuning datasets were selected to be meaningfully out-of-distribution, not only in flow regime but also in including thermal and electromagnetic components that are not represented at all in the pretraining data. Training was performed on the Frontier and Perlmutter supercomputers at the Oak Ridge Leadership Computing Facility (OLCF) and National Energy Research Scientific Computing Center (NERSC), respectively, using distributed data parallelism.

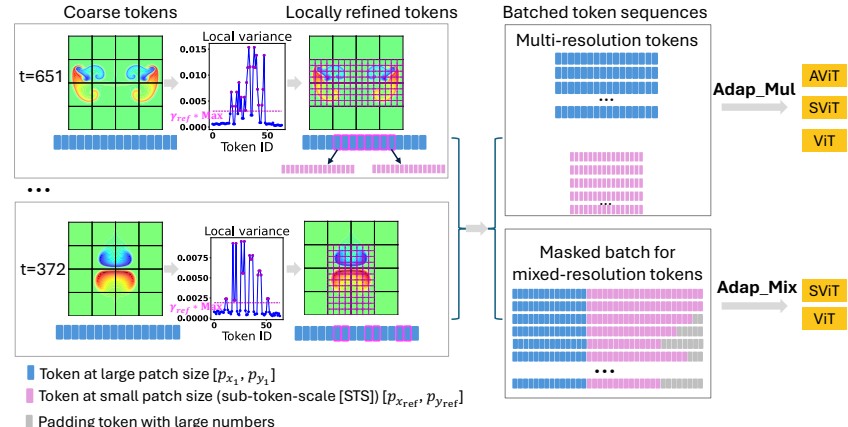

Figure 2: Adaptive tokenization that dynamically adjusts patch sizes based on local features. There are three essential parameters: $[p_{x_1}, p_{y_1}]$, $[p_{x_{\mathrm{ref}}}, p_{y_{\mathrm{ref}}}]$ and $\gamma_{\mathrm{ref}}$. The parameter $[p_{x_1}, p_{y_1}]$ denotes the coarse patch size to start with, $[p_{x_{\mathrm{ref}}}, p_{y_{\mathrm{ref}}}]$ represents the refined patch size, and $\gamma_{\mathrm{ref}} \in [0, 1]$ determines which patches to refine. We select patches with local variances greater than $\gamma_{\mathrm{ref}}$ times the maximum variance across all patches.

## 4 EXPERIMENTS

We design three experiments to evaluate 1) the performance of three spatiotemporal attention schemes, AViT, SViT, and ViT, 2) the impact of adaptive tokenization, and 3) the effectiveness of pretrained models on two fine-tuning tasks that feature physics different from the pretraining data.

### 4.1 SPATIOTEMPORAL ATTENTION SCHEMES

We evaluate AViT, SViT, and ViT for three model sizes: Tiny (Ti), Small (S), and Base (B) with 3, 6, 12 heads and hidden dimension $C_{\mathrm{emb}} = 192, 384$, and $768$, respectively (Touvron et al., 2022), as shown in Table 1, on the colliding thermals dataset. In the same size category, AViT and SViT are about 30% larger than ViT due to the additional attention block. More details about the experiment are presented in Appendix A.2.

Table 1: Number of model parameters in AViT, SViT, and ViT for three model sizes, Tiny, Small, and Base, detailed in Section 4.1. ViT results in about 30% fewer model parameters than AViT and SViT because the latter two require additional attention blocks.

|      | Tiny | Small | Base |
|------|------|-------|------|
| AViT | 7.5M | 29.9M | 119.3M |
| SViT | 7.6M | 30.0M | 119.3M |
| ViT  | 5.8M | 22.8M | 90.9M |

Figure 3 compares the final test error, defined as the normalized root-mean-square error (NRMSE), and the training time, represented as GPU hour per step, for the nine models. For the same size category, SViT (green) achieves the lowest error, followed by ViT (blue), and then AViT (red). In terms of training time, SViT takes longer than AViT, while ViT is the least expensive one. ViT processes longer token sequences and hence is expected to have a higher single-unit attention cost, whereas AViT and SViT have multiple attention units with shorter token sequence length. The results reported in Figure 3 show that the ViT has the lowest cost, which implies that the number of attention blocks plays a more important role than the token sequence length in terms of training cost in this example. This observation is due to the fact that the spatiotemporal token sequence length ($16 \times 8 \times 8$) in this example is relatively short. We expect ViT to become more expensive than AViT and SViT when more refined or higher dimensional solutions are considered, in which longer token sequences are required. In general, we find that SViTs and ViTs are more expressive

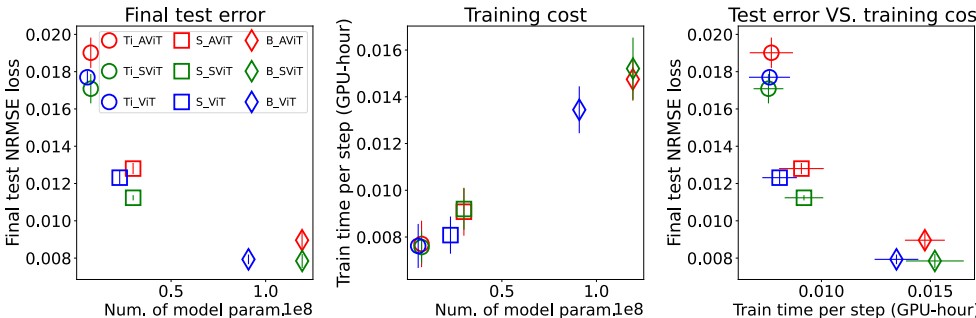

Figure 3: Learning efficiency of AViT, SViT, and ViT at three model sizes regarding final predictive error and training time cost: SViT and ViT are observed to be more expressive and computationally efficient than AViT in the experiment, as they require fewer model parameters and less training time to achieve the same test accuracy.

and computationally efficient than AViTs, in that they achieve the lower predictive errors with fewer model parameters and less training time.

## 4.2 ADAPTIVE TOKENIZATION

We start the evaluation of our adaptive tokenization methods in a single collision trajectory between two thermal bubbles. Figure 4 compare the temperature contours of the true solution at $t = 590$ with the predicted solutions from Ti-AViT models at constant patch sizes: ps=$16 \times 16$ and ps=$32 \times 32$ and adaptive tokenization (Adap_Mul with $p_{x_1} = p_{y_1} = 32$, $p_{x_{\text{ref}}} = p_{y_{\text{ref}}} = 16$ , and $\gamma_{\text{ref}} = 0.2$). The predicted solution from ps=$32 \times 32$ exhibits abrupt changes with clear edges for the local structures inside the patches, while the finer resolution model at ps=$16 \times 16$ captures smoother, finer structures but requiring more patches. In contrast, our adaptive tokenization method (Adap_Mul) capture smooth, fine structures comparable to ps=$16 \times 16$ while requiring much fewer FLOPs, as shown in Figure 5.

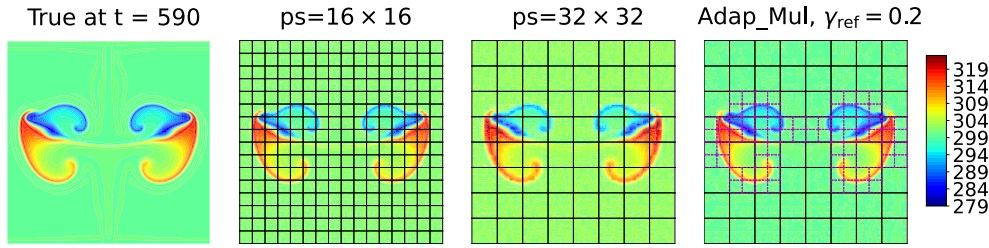

Figure 4: Predicted temperature contours at $t = 590$ from Ti-SViT models with constant patch sizes ps=$16 \times 16$ and ps=$32 \times 32$ and adaptive tokenization (Adap_Mul with $p_{x_1} = p_{y_1} = 32$, $p_{x_{\text{ref}}} = p_{y_{\text{ref}}} = 16$ , and $\gamma_{\text{ref}} = 0.2$). Adap_Mul predicts smoother, finer local structures that are overlooked in ps=$32 \times 32$, similar to the more expensive ps=$16 \times 16$.

Figure 5 shows the final NRMSE loss versus the floating-point operations (measured in TeraFLOPs) for 15 models, including the three in Figure 4. We aim to evaluate the adaptive tokenization methods coupled with (Ti-) AViT, SViT, and ViT versus using these models with three fixed patch resolutions: patch sizes ps=$8 \times 8$, $16 \times 16$, and $32 \times 32$. For the same attention scheme with a fixed patch size, as expected, increasing resolution leads to lower errors but also substantially increases the training cost, particularly for ViT (triangles). ViT shows fewer FLOPs than AViT (squares) and SViT (circles) with shorter sequences (blue), consistent with the time measure in Figure 3, but it significantly surpasses the other two at the finest resolution with longer sequences (red). In contrast, our adaptive scheme (magenta markers, Adap_Mul with $p_{x_{\text{ref}}} = 16$), which starts with uniform $32 \times 32$ patches and locally refines to $16 \times 16$ on selected patches, achieves comparable accuracy to uniform $16 \times 16$ patches with SViT and ViT. Moreover, Adap_Mul with $p_{x_{\text{ref}}} = 16$ obtains this accuracy level at reduced FLOPs. As this reduced cost depends significantly on the spatiotemporal attention, the speedup is modest for

SViT but becomes more significant for ViT, being more than $2\times$ more efficient than constant patch sizes for ViT.

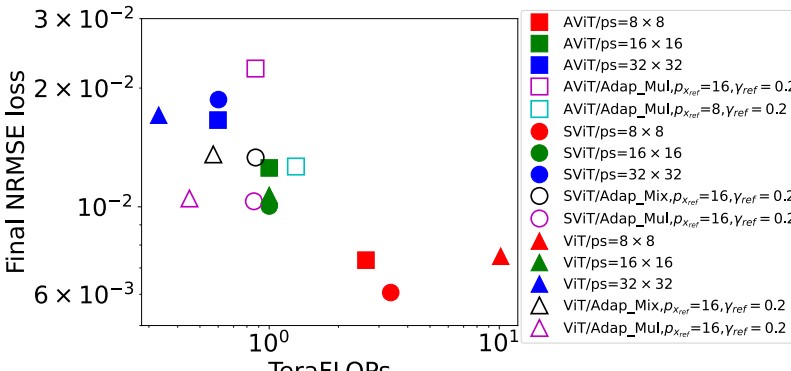

Figure 5: Final NRMSE loss for the three attention schemes (AViT, SViT, and ViT) with adaptive tokenization and constant patch sizes against estimated TeraFLOPs at the Tiny size. The flops values are available in Table A1. Adaptive tokenization methods (empty markers, $p_{x_1} = 32$) achieves comparable accuracy to ps=$16 \times 16$ (green solid markers) but with $2.2\times$ fewer FLOPs in ViT (triangles) and $1.16\times$ fewer FLOPs in SViT (circles), respectively. In contrast, AViT (squares), under the same setting, achieves higher error (magenta square with $p_{x_{\mathrm{ref}}} = 16$) or lower error (cyan square with $p_{x_{\mathrm{ref}}} = 8$) but with refined patches, making it less suitable for adaptive tokenization.

While the results with adaptive toknenization are positive on a single trajectory from the colliding thermals dataset, the accuracy improvement deteriorates when applied to more complex settings with multiple trajectories that invovle varying initial bubble locations and temperature differences. Figure 6 compares the final test errors of Ti-SViT with constant patch sizes: ps=$32 \times 32$, ps=$16 \times 16$, and adaptive tokenization with a few parametric settings. In general, the adaptive errors are between errors of the two reference cases but still noticeably higher than the error from ps=$16 \times 16$. The adaptive error decreases with lower $\gamma_{\mathrm{ref}}$ values and is ideally expected to converge to the model error with the constant fine patch size. However, we encounter training instability issues when further reducing $\gamma_{\mathrm{ref}}$. Addressing these stability issues is a focus of future work.

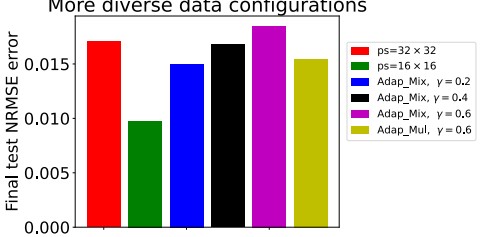

Figure 6: Final NRMSE test loss from constant patch sizes (ps=$32 \times 32$ and ps=$16 \times 16$) and adaptive tokenization for diverse data configurations with initial bubble locations and intensities varying dramatically. Adpative tokenization (with $p_{x_1} = 32$, $p_{x_{\mathrm{ref}}} = 8$) shows varying accuracy improvement with different parameter settings but the improvement is not as optimal as the single trajectory case in Figure 5.

### 4.3 EFFECTIVENESS OF PRETRAINING IN COLLIDING THERMALS AND MHD FINE-TUNING TASKS

We examine the transferrability of pretrained models to fine-tuning systems with distinct physics and different set of variables, as in Table A2. Specifically, we aim to address three broad questions:

1. Is pretraining effective when the downstream tasks have a distinct set of physical variables?

2. How does limited fine-tuning of non-attention blocks compare to full fine-tuning?

3. How does fine-tuning data size affect convergence?

To address these three questions, we design a sets of experiments, starting from models pretrained on PDEBench or randomly initialized models ('*_INIT'), and fine-tune them on colliding thermals and MHD datasets with distinct physical variables. For fine-tuning each model, we either allow all model parameters to be tunable ('ALL') or freeze the attention blocks and limit training to the preprocessor, the tokenization module, and the postprocessor ('PREPOST'). Finally, for each initial model and fine-tuning configuration, we train four models with increasing amounts of fine-tuning data.

For the colliding thermals dataset, Figure 7 compares the test loss with full and limited fine-tuning using pretrained and randomly initialized models. The different training data sizes ranging from one set of colliding thermals time-trajectory to 24 sets of trajectories. The fine-tuning task is to predict the solution of the physical system at a lead time of $t_{\text{lead}}$ uniformly sampled between 1 and 50 steps. An example of the true and predicted solutions in these four training configurations is illustrated in Figure 8.

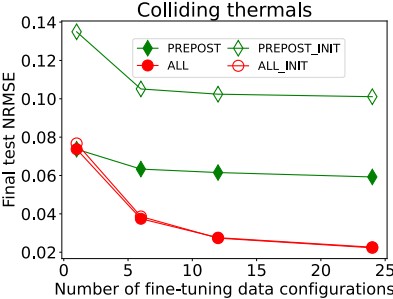

Figure 7: NRMSE loss for test set at different training data sizes in fine-tuning of colliding thermals at a maximum lead time of 50 steps, with full ('ALL') and limited ('PREPOST') fine-tuning using pretrained and randomly initialized models ('*_INIT').

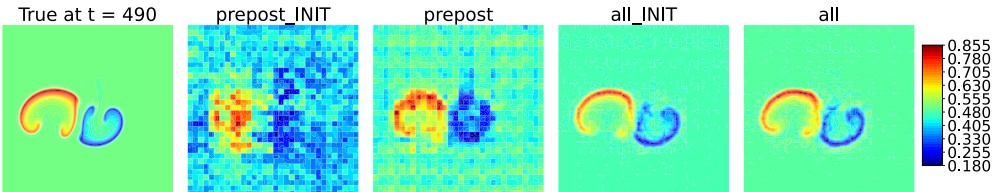

Figure 8: Temperature contours of true solution vs predicted solutions from four fine-tuned models (on 12 trajectories) at $t = 490$ from Ti-SViT models for a lead time of 40 in the collision of two thermal bubbles.

For the limited fine-tuning test with the colliding thermals dataset, the pretrained models achieve significantly lower error than starting from scratch with randomly initialized parameters. Moreover, while this advantage persists as the number of fine-tuning data increases, it is most pronounced in the low data configuration of learning from a single trajectory. Indeed, we find that limited fine-tuning with the pretrained models generalizes well even when learning from one trajectory, seeing only moderate improvements when run on the largest dataset size considered. Overall, the lower converged error from pretrained models suggests the frozen attention blocks clearly learned transferable knowledge during pretraining. For full fine-tuning, the accuracy is much better than limited fine-tuning as a result of the model being more expressive. The difference between the pretrained and randomly initialized models is much lower, being minor in the case of a single data configuration during training and vanishing as the amount of data increases.

For the MHD dataset, Figure 9 shows the final test NRMSE errors in lid-driven cavity flows after fine-tuning against data sizes when starting from pretrained and randomly initialized models for

limited and full fine-tuning. The training dataset sizes used for fine-tuning range from 1 to 12 simulation configurations, with each configuration including approximately 1900 samples. The fine-tuning task is to predict the flow solution at a lead time of $t_{\text{lead}}$ uniformly sampled between 1 and 100 steps. Contour plots from the true solution and the predicted solution from each training configuration are depicted in Figure 10.

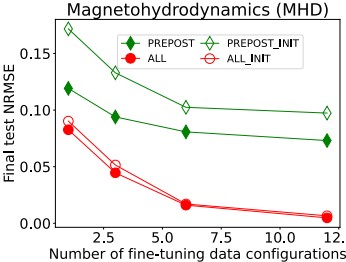

Figure 9: NRMSE loss for test set at different training data sizes in fine-tuning of lid-driven cavity MHD flows dataset at a maximum lead time of 100 steps, with full ('ALL') and limited ('PREPOST') fine-tuning using pretrained and randomly initialized models ('*_INIT').

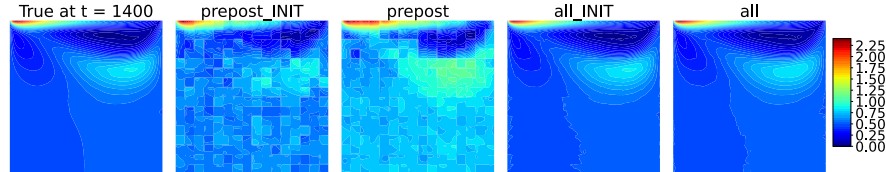

Figure 10: Contours of true horizontal magnetic field values $B_x$ vs predicted solutions from four fine-tuned models (on 12 trajectories) at $t = 1400$ from Ti-SViT models for a lead time of 80 in lid-driven cavity MHD flows.

Overall, the fine-tuning performance is a result of model expressibility, training data size, and the similarity between training and testing tasks. As with the colliding thermals dataset, pretrained models outperformed the randomly initialized models for both full and limited fine-tuning runs. However, the reduced expressibility of the limited fine-tuning configuration consistently shows an accuracy gap, even with more training data, as they cannot fully represent the data complexity. In contrast, full fine-tuning leads to more expressive models that can capture all training data information when trained on limited data but often show high test errors; as more training data is provided, they generalize well and lead to a convergent improved test error. In our fine-tuning, the randomly initialized models perform well in testing even with a single data configuration, likely due to the similarity between training and testing tasks. Future work will explore more challenging scenarios with increased heterogeneity within the fine-tuning data.

While studies like McCabe et al. (2023) have demonstrated impressive outperformance from fine-tuning of pretrained models versus randomly initialized models, these fine-tuning tests were performed on data that, while distinct, was fully governed by physical equations and characterized by physical variables that were represented in the training data. Yet for a model that aims to be foundational for multiphysical systems, we argue that assessing model performance in more realistic settings, where equations like Navier-Stokes are coupled with those from other domains of physics, is a more informative test of the effectiveness of pretraining. Accordingly, we assess fine-tuning performance on physical systems that incorporate fluid flows, which are well-represented in PDEBench, with thermodynamics and electromagnetism, which are not. As reasonably anticipated, we find that advantages of pretraining are reduced in this more complex setting.

## 5 DISCUSSION

In this paper, we make three contributions that will advance the development of foundation models for multiscale physical systems. First, we find that while some data efficiency is lost in a fully

decoupled spatiotemporal attention scheme, such as AViT and SViT, provides an intriguing balance of computational and data efficiency versus the standard ViT approach. Yet using SViT alone does not sufficiently address the computational challenges associated with attention for high spatial resolutions. Second, we instead suggest that our adaptive tokenization scheme provides a promising approach for working with high resolution data. This sort of adaptivity has the potential to be both flexible and expressive enough to deal with the dynamic and sparse nature of the multiscale features in physical data. Third, we suggest an alternative path to evaluate foundation models for multiscale physical systems that focuses on fine-tuning problems involving out-of-distribution physics governed by different equations with distinct sets of physical variables. In two such settings, colliding thermals and magnetohydrodynamics, we find that while pretraining does provide an advantage, its impact is much more muted compared to fine-tuning on the same set of variables, suggesting additional effort is required to obtain truly foundational models in this space.

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
