# A APPENDIX

## A.1 DATASETS

Three datasets were used in the work: PDEBench (Takamoto et al., 2022), colliding thermals (Norman, 2024), and lid-driven cavity MHD flows.

- PDEBench (https://github.com/pdebench/PDEBench) consists of diverse 1D, 2D, and 3D diverse benchmark datasets. We used the 2D cases – incompressible flows, compressible flows, turbulent flows, reaction diffusion, and shallow water – for model pre-training in Section 4.3. The govern equations are summarized below.

  – Shallow water equations [swe]:

  $$\partial_t h + \nabla \cdot (h\boldsymbol{v}) = 0,$$

  $$\partial_t(h\boldsymbol{v}) + \nabla \cdot \left(\frac{1}{2}h\boldsymbol{v}^2 + \frac{1}{2}g_r h^2\right) = -g_r h \nabla b$$

  – Diffusion-reaction equations [diffre2d]:

  $$\partial_t \boldsymbol{c} = \boldsymbol{D}\nabla^2 \boldsymbol{c} + \boldsymbol{R}(\boldsymbol{c}),.$$

  where $\xi$ and $\phi$ in $\boldsymbol{c} = [\xi, \phi]$ are the activator and the inhibitor, respectively.

  – Incompressible NS [Incomp]:

  $$\nabla \cdot \boldsymbol{v} = 0,$$
  $$\rho\left(\partial_t \boldsymbol{v} + \boldsymbol{v} \cdot \nabla \boldsymbol{v}\right) = -\nabla p + \eta \nabla^2 \boldsymbol{v} + \boldsymbol{f}$$

  – Compressible NS [compNS] with random and turbulent initial conditions:

  $$\partial_t \rho + \nabla \cdot (\rho \boldsymbol{v}) = 0,$$

  $$\rho\left(\partial_t \boldsymbol{v} + \boldsymbol{v} \cdot \nabla \boldsymbol{v}\right) = -\nabla p + \eta \nabla^2 \boldsymbol{v} + (\zeta + \eta/3)\nabla(\nabla \cdot \boldsymbol{v})$$

  $$\partial_t \left[\epsilon + \frac{\rho \boldsymbol{v}^2}{2}\right] + \nabla \cdot \left[\left(\epsilon + p + \frac{\rho \boldsymbol{v}^2}{2}\right)\boldsymbol{v} - \boldsymbol{v} \cdot \boldsymbol{\sigma}'\right] = 0$$

  with $\epsilon = p/\Gamma - 1$ and $\Gamma = 5/3$.

  For more details on these cases and equations, users are referred to (Takamoto et al., 2022).

- The colliding thermals dataset was generated for our work, and the details will be presented in Section A.1.1. It was used in the experiments in Sections 4.1 and 4.2, and also as one of the two fine-tuning cases in Section 4.3.

- Lid-driven cavity MHD dataset was also generated in our work, and it was used as the other fine-tuning case in Section 4.3. We will present the details in Section A.1.2.

## A.1.1 COLLIDING THERMALS

Thermal collision datasets contains multiple time history trajectories of the mixing of two bubbles- one cold bubble at the top colliding with a warm bubble at the bottom. Details about the governing equations can be found in Norman (2024). These trajectories start from different initial temperature conditions as

$$T_0(x, z) = 300.0 + T_{10}(x, z) + T_{20}(x, z), \tag{2}$$

with one hot $T_{10}$ and cold $T_{20}$ thermals being

$$T_{10}(x, z) = \begin{cases} Tc_1 \cos\left(\frac{\pi}{2}d_1(x, z)\right)^2, & \text{if } d_1(x, z) \leq 1 \\ 0, & \text{otherwise} \end{cases} \tag{3}$$

and

$$T_{20}(x, z) = \begin{cases} -Tc_2 \cos\left(\frac{\pi}{2}d_2(x, z)\right)^2, & \text{if } d_2(x, z) \leq 1 \\ 0, & \text{otherwise} \end{cases} \tag{4}$$

where $Tc_i$ is the center temperature amplitude and $d_i(x, z) = \sqrt{\frac{(x-xc_i)^2}{rx_i^2} + \frac{(z-zc_i)^2}{rz_i^2}}$ is the distance from thermal center $(xc_i, zc_i)$ for $i = 1, 2$. The thermals are elliptical in shape with the radius, $rx_i$ and $rz_i$, in x and z directions, respectively.

**Configurations** We sample 4096 configurations with the thermals ($i = 1, 2$) at different locations following uniform distribution,

$$xc_i \sim U[0.2L, 0.8L], \, zc_1 \sim U[0.2L, 0.3L], \text{ and } zc_2 \sim U[0.7L, 0.8L], \tag{5}$$

with different elliptical shapes also following uniform distribution,

$$rx_i \sim U[0.1L, 0.2L] \text{ and } rz_i \sim U[0.1L, 0.2L], \tag{6}$$

and with temperature amplitudes equally sampled from,

$$Tc_i \sim C\{10, 15, 20, 25\}. \tag{7}$$

The equations are solved by using a finite volume method with $nx = 256, ny = 256$ grid points in x and z directions, respectively. The simulations are advanced in time for 500 seconds and solutions are saved every 0.5 second. In total, we have 4096 trajectories, each with data at size ($nt = 1001, nx = 256, ny = 256$).

### A.1.2 LID-DRIVEN CAVITY MAGNETOHYDRODYNAMICS (MHD) FLOWS

The MHD dataset contains solution trajectories from initial conditions to steady states for a benchmark lid-driven cavity MHD flow problem in two dimensions with varying configurations. The MHD flow is governed by an incompressible Navier-Stokes equation with Lorentz force coupled with an induction equation with divergence cleaning. The detail formulation of the governing equations and problem setting for the lid-driven MHD cavity problem are given in Fambri et al. (2023).

**Configurations** In this dataset, we include solution trajectories of the lid-driven cavity problem at three magnetic Reynolds numbers $Re_m = 100, 200$, and $500$, each with ten external horizontal magnetic field magnitude $B_x = 0.05, 0.10, \ldots, 0.50$. This gives 30 different problem configurations. For each problem configuration, the fluid velocity field $\mathbf{v}$ and the magnetic field $\mathbf{B}$ are recorded on a $128 \times 128$ uniform spatial mesh for 2,000 time steps.

### A.2 MORE ON SPATIOTEMPORAL ATTENTIONS AND ADAPTIVE TOKENIZATION

**Training setting** We randomly sampled a subset with 512 trajectories for training and 64 trajectories for testing for the results in Sections 4.1 and 4.2. During training, we use the `AdamW` optimizer with a learning rate equal to $10^{-4}$. Batch size was set to be 128 and accumulate gradient step was set to be 1. Models were trained for 20,000 steps. For cases with constant patch size, the value was set to be $32 \times 32$.

For the experiment on spatotemporal attention schemes in Section 4.1, we ran 9 cases with AViT, SViT, and ViT attention blocks at three sizes (Ti, S, and B). Figure A1 shows the loss history during training of the models for both training and test sets, and Figure A2 shows the training time cost.

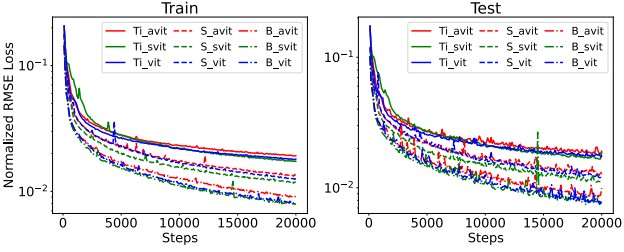

Figure A1: Loss history of three spatiotemporal attention schemes at three model sizes during training

For the experiment on adaptive tokenization in Section 4.2, Figure A3 shows the training loss of all models in a single colliding thermal trajectory. Table A1 shows the estimated TeraFLOPS values for the models in Figure 5. Figure A4 shows the loss history of models with adaptive tokenization and constant patch sizes during training on more diverse colliding thermals dataset.

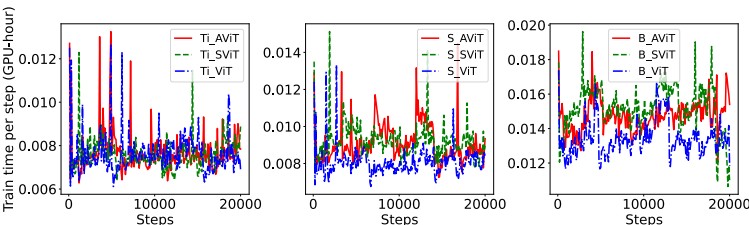

Figure A2: Training time per step of three spatiotemporal attention schemes at three model sizes

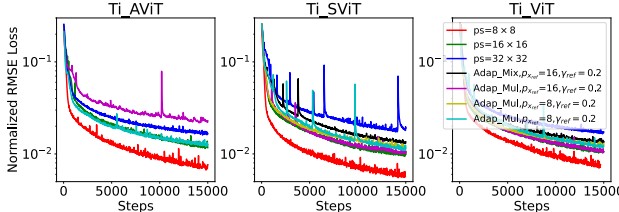

Figure A3: Comparison of training loss histories of models with adaptive tokenization and constant patch sizes (ps=$32 \times 32$, ps=$16 \times 16$, and ps=$8 \times 8$) for the three spatiotemporal attention schemes (AViT, SViT, and ViT) in a single colliding thermals trajectory.

Table A1: Estimated TeraFLOPs for results in Figure 5 for $T = 16$

| | ps= $32 \times 32$ | ps= $16 \times 16$ | ps= $8 \times 8$ | Adap_Mul, $p_{x_{\mathrm{ref}}} = 16$, $\gamma_{\mathrm{ref}} = 0.2$ | Adap_Mix, $p_{x_{\mathrm{ref}}} = 16$, $\gamma_{\mathrm{ref}} = 0.2$ | Adap_Mul, $p_{x_{\mathrm{ref}}} = 8$ $\gamma_{\mathrm{ref}} = 0.2$ |
|---|---|---|---|---|---|---|
| Ti-AViT | 0.088 | 0.147 | 0.386 | 0.128 | - | 0.192 |
| Ti-SViT | 0.085 | 0.141 | 0.475 | 0.121 | 0.123 | - |
| Ti-ViT | 0.09 | 0.272 | 2.76 | 0.122 | 0.155 | - |

## A.3 PRETRAINING AND FINE-TUNING

### A.3.1 PRETRAINING

Five 2D datasets from PDEBench Takamoto et al. (2022) were used for pretraining, including shallow water, diffusion reaction, incompressible flows, compressible flows, and turbulent flows. The details of these datasets including physical variables, spatiotemporal resolutions, and number of trajectories are summarized in Table A2.

During training, we used the `AdamW` optimizer with `DAdaptAdam` for learning rate scheduling. Batch size was set to be 1472 and patch size was $32 \times 32$. Training/testing/validating split was 0.8/0.1/0.1. Gradient accumulation was set to be 1. We trained the model for 30,000 steps to predict the next step solution given a history of $T = 16$.

### A.3.2 FINE-TUNING

For fine-tuning, we evaluate the transferrability of pretrained models to systems with distinct physics and different sets of variables. Table A2 summarizes the two fine-tuning cases: colliding thermals and lid-driven cavity MHD flows. In the two cases, pretrained models were fine-tuned to predict the solution at a future time $t + t_{\mathrm{lead}}$ given a history of solutions from $t - T + 1$ to $t$. In our experiments, $T$ was set to be 10 while $t_{\mathrm{lead}}$ was set to 50 for the colliding thermals and 100 for the lid-driven cavity MHD flows. The fine-tuned models were evaluated on a held-out test set for all runs in each case. We used the `AdamW` optimizer with a learning rate equal to $10^{-4}$. Batch size was set to be 256. Models were fine-tuned for 600 epochs for colliding thermals and 1000 epochs for lid=drive cavity MHD flows.

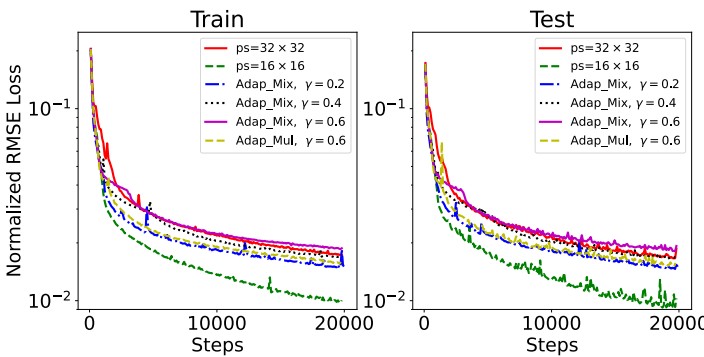

Figure A4: Comparison of training loss histories of models with adaptive tokenization and constant patch sizes (ps=$32 \times 32$ and ps=$16 \times 16$) for Ti-SViT in 512 colliding thermals trajectories.

Table A2: Cases and datasets

**Pretraining**: PDEBench Takamoto et al. (2022)

| Dataset | Variables ($C$) | Spatiotemporal res. ($T \times H \times W$) | $N_{\text{traj}}$ trajectories |
|---|---|---|---|
| Shallow-water | $h$ | $101 \times 128 \times 128$ | 1,000 |
| Diffusion-reaction [diffre2d] | $\xi, \phi$ | $101 \times 128 \times 128$ | 1,000 |
| Incompressible NS | $u, v, \rho_{\text{aug}}$ | $1000 \times 512 \times 512$ | 992 |
| Compressible NS Rand-128 | $u, v, \rho, P$ | $21 \times 128 \times 128$ | 40,000 |
| Compressible NS Rand-512 | $u, v, \rho, P$ | $21 \times 512 \times 512$ | 2,000 |
| Compressible NS Turb | $u, v, \rho, P$ | $21 \times 512 \times 512$ | 2,000 |

**Fine-tuning**: colliding thermals (Section A.1.1) and lid-driven MHD (Section A.1.2)

| Dataset | Variables ($C$) | Spatiotemporal res. ($T \times H \times W$) | $N_{\text{traj}}$ trajectories in training |
|---|---|---|---|
| colliding thermals | $\rho, u, v, T$ | $1001 \times 256 \times 256$ | [1, 6, 12, 24, 48] |
| lid-driven MHD | $u, v, B_x, B_y$ | $2000 \times 128 \times 128$ | [1, 3, 6, 12, 24] |

**Colliding thermals**    We sampled 1, 6, 12, and 24 trajectories for training. The results in Section 4.3 are shown for a fixed test set with 24 trajectories.

**Lid-driven cavity MHD flows**    Among the 30 cases, we kept 6 for testing. From the remaining 24 cases, we sampled 1, 3, 6, and 12 cases to assess the impact of the amount of fine-tuning data.