# OpenReview forum: "MATEY: multiscale adaptive foundation models for spatiotemporal physical systems"
_ICLR.cc/2025/Conference — ICLR 2025 Conference Withdrawn Submission_

### Official Review · Reviewer_dtxa · 2024-10-28

**Soundness:** 2
**Presentation:** 2
**Contribution:** 2
**Rating:** 3
**Confidence:** 3

**Summary:**

- The authors propose MATEY, a foundation model for spatiotemporal physical systems;
- MATEY adopts various spatiotemporal attention module for implementation;
- MATEY proposes a local patch variance based adaptive tokenization algorithm for patch sequence refinement;
- MATEY is trained on PDEBench data and finetuned on two cases.

**Strengths:**

- Transferred AViT/SViT/ViT from computer vision area to spatiotemporal physical system prediction area;
- Designed an adaptive tokenization algorithm for patch sequence refinement;
- Carried out extensive ablation study on the model structures.

**Weaknesses:**

- The AViT/SViT/ViT modules implemented in this paper come directly from previous work, which **limits the novelty on MATEY's network structure**;
- **Lack of details** for the proposed adaptive tokenization mechanism. The pipeline for adaptive tokenization remains unclear. Further illustration on the local patch variance threshhold and the densifying mechanism (refered as STS in the paper) are needed, e.g., how are the patches splited into sub-tokens;
- **Baselines missing**. There should be at least one baseline, no matter AI-based or simulation-based, for 1) colliding thermals between a
cold and a warm bubbles from MiniWeather simulations and 2) lid-driven cavityMHD flows, since these are classic questions for spatiotemporal physical systems simulation. However, this kind of comparision is missing throughout all the experiment part, which makes it extremely difficult to determine how effective MATEY really is.

**Questions:**

- About the model part: 1) how are the coarse patches splited into sub-tokens? 2) What's the criteria for choosing γref, and how much will it affect the GPU memory occupation and predict accuracy?
- About the experiment: 1) How much performance gain on the two downstream tasks can MATEY achieve compared to previous method? 2) How much will the input resolution affect the model performance? Do you have data showing, or are there any existing results indicating that input resolution affects the model performance?

**Details Of Ethics Concerns:**

On Line 214/215, page 4, it is mentioned that training was performed on the Frontier and Perlmutter supercomputers at the Oak Ridge Leadership Computing Facility (OLCF) and National Energy Research Scientific Computing Center (NERSC). This might be against the double-blind review requestments.

---

### Official Review · Reviewer_fEN8 · 2024-11-04

**Soundness:** 2
**Presentation:** 2
**Contribution:** 1
**Rating:** 3
**Confidence:** 4

**Summary:**

This paper studies scale-adaptive tokenization in vision transformer models.  Vanilla ViTs compute dense pair-wise attention of all input tokens, resulting in poor efficiency for long-horizon or high-resolution spatio-temporal data.  To reduce the computational overhead, existing solutions either hierarchically merge finer tokens into coarser tokens or modulate the receptive field of attention operation.  Inspired by the former type of approaches, this paper proposes a method that adaptively selects the patch size based on local feature statistics, where smaller patches are used to extract tokens in high-variance regions.  This paper demonstrates better trade-off between efficiency and performance with the proposed method on PDEBench dataset.

**Strengths:**

1. The paper is written well.  The high-level idea and the context of the problem is discussed clearly in the introduction.

2. The idea of using scale-adaptive patchification is interesting.

**Weaknesses:**

1. Even though the paper is written well, the experimental results are disappointing. According to Figure 6, the scale-adaptive patchification doesn't bring performance gain, compared to the baseline with constant-size patchification.  According to Figure 7 and 9, the proposed method does not improve the performance in the transfer learning setup.

2. The proposed method may not be as efficient and effective as described in the paper.  The model not only needs more tokens (multi-scale tokens), but also additional iteration steps to break down the patch size.  Especially, the selection of patch size depends on the feature statistics, which requires to extract features multiple times.  The paper does not provide any analysis on the inference latency of the model.

3. The paper does not discuss any state-of-the-art sequential models (e.g. MambaVision[1]), which is as effective but more efficient than transformers.

---

Reference:

[1] MambaVision: A Hybrid Mamba-Transformer Vision Backbone.  Ali Hatamizadeh and Jan Kautz.

**Questions:**

1. What is the throughput (images / second) of the proposed method, compared to the baseline?

2. Following Q1, What is the latency of each component?

3. What is the performance of Mamba models on PDEBench?

---

### Official Review · Reviewer_Jysn · 2024-11-04

**Soundness:** 2
**Presentation:** 2
**Contribution:** 2
**Rating:** 3
**Confidence:** 3

**Summary:**

In this paper the main focus is on three aspects - one to propose an adaptive tokenization scheme, second - to benchmark various attention mechanisms - decomposed spatial vs temporal attention mechanisms, third to understand the generalization of pretrained models on out of distribution phenomenon. The focus is on obtaining insights to aid development of foundation models for physical systems.

**Strengths:**

* The analysis by the authors suggests that adaptive tokenization is useful in certain scenarios - that is, the results are positive on a single trajectory from the colliding thermals dataset, the accuracy improvement deteriorates when applied to more complex settings with multiple trajectories that involve varying initial bubble locations and temperature differences.
* The utility of decoupling attention through S-ViT and A-Vit demonstrates that these decoupled attention mechanisms are not yet conclusively better than ViTs
* The pretraining on PDEBench and evaluation on colliding thermals and magnetohydrodynamics, demonstrates reduced advantages as compared to the previous study by McCabe et al.

**Weaknesses:**

* While the analysis by the authors is helpful towards improved training of foundation models, it is not clear that a definite strategy towards this goal is identified or proposed in this paper.
* The adaptive tokenization proposed here is useful, however, it is not benchmarked against other adaptive tokenization mechanisms for instance the prune and merge strategy from Haurum et al or Meng et al.
* The work does not provide a definitive solution towards an effective attention mechanism. We do see a comparison based on A-Vit and S-Vit. Currently there is also interest in exploring these mechanisms using neural operators such as Adaptive Fourier Neural Operators. It is not clear whether there exists a better attention mechanisms to obtain high-quality multiscale solutions without considering such operators that are not considered at all in this study.
* The fine-tuning results are useful. But, it would also be useful to consider in conjunction with similar studies such as McCabe et al 2023 to validate that the study is done correctly - that is the fine-tuning does work for fluid flow based tasks but is limited for more complex tasks.

**Questions:**

The clarification questions would be whether clarifications can be provided with respect to the weaknesses mentioned above

---

### Official Review · Reviewer_cX2d · 2024-11-07

**Soundness:** 2
**Presentation:** 3
**Contribution:** 2
**Rating:** 5
**Confidence:** 3

**Summary:**

In this paper, the authors propose adaptive tokenization that could set different token sizes based on different local physical features, and decouple the spatial and temporal dimensions to do attention for relatively long-range sequences to solve physical representations. The authors have evaluated the proposed method on the colliding thermals dataset. Further experiments of fine-tuning the pretrained models on the different physical properties show the generalization and expressibility of the pretrained model.

**Strengths:**

- It is interesting to apply adaptive tokenization to attention, especially with long sequences. This could also be applied to problems other than physical science.
- The paper is easy to understand and has relatively well-organized arguments.
- The analysis and the experiments on different physical problems show good performance, and the generalization of the proposed method.

**Weaknesses:**

- How does this adaptive tokenization compare to simple coarse-to-fine tokenization? Some ablation studies could be conducted to verify the necessity of adaptive tokenization.
- The proposed method does not seem to have an advantage in computation, including time and memory consumption compared to the recent arts.
- Although the authors have shown that the pretrained model showed better performance with limited fine-tuning with quantitative results, there are no relatively large statistical results showing the benefits of the pretrained model.
- According to the paper, the proposed method should be able to handle long-range correlations, which refer to long sequences. However, the experimented sequence length is still limited, which I think might not be very suitable to emphasize the advantage of the proposed method.
- Also, the authors could provide the results of even finer path size to compare the proposed method with “definite good performance” to highlight the effectiveness of the adaptive tokenization.
- What is the metric to decide the choice of the patch size? The authors mentioned local physical features, but how specifically do the authors form this problem? It could be made clearer in the method section.
- Naming the model foundation model for spatiotemporal physical systems, however, the experiments and the datasets are limited.

**Questions:**

I think the idea of adaptive tokenization is interesting and can be applied to many other fields. However, I think the authors did not emphasize the importance and the generalization of the application of the proposed tokenization, which is, according to the paper, the most important part that contributed to the performance of the proposed method.
Also, experiments and analysis should focus on long sequences and large-scale datasets.

Please see my detailed comments above.

---

### Note · Authors · 2025-01-02

**Comment:**

We appreciate the reviewers' feedback. After careful consideration, we have decided to withdraw our paper from the current ICLR review cycle. We plan to further develop and improve it based on the feedback and submit the revised manuscript to another venue.

**Withdrawal Confirmation:**

I have read and agree with the venue's withdrawal policy on behalf of myself and my co-authors.